# Differences between Arterial and Venous Umbilical Cord Plasma Metabolome and Association with Parity

**DOI:** 10.3390/metabo12020175

**Published:** 2022-02-13

**Authors:** Olle Hartvigsson, Malin Barman, Otto Savolainen, Alastair B. Ross, Anna Sandin, Bo Jacobsson, Agnes E. Wold, Ann-Sofie Sandberg, Carl Brunius

**Affiliations:** 1Division of Food and Nutrition Science, Department of Biology and Biological Engineering, Chalmers University of Technology, 412 96 Göteborg, Sweden; malin.barman@chalmers.se (M.B.); ann-sofie.sandberg@chalmers.se (A.-S.S.); carl.brunius@chalmers.se (C.B.); 2Institute of Environmental Medicine, Karolinska Institutet, 171 77 Stockholm, Sweden; 3Chalmers Mass Spectrometry Infrastructure, Chalmers University of Technology, 412 96 Göteborg, Sweden; otto.savolainen@chalmers.se; 4AgResearch, Proteins and Metabolites, Lincoln 7674, New Zealand; alastair.ross@agresearch.co.nz; 5Department of Clinical Science, Pediatrics, Sunderby Research Unit, Umeå University, 901 897 Umeå, Sweden; anna.sandin@umu.se; 6Department of Obstetrics and Gynecology, Institute of Clinical Sciences, Sahlgrenska Academy, University of Gothenburg, 413 45 Gothenburg, Sweden; bo.jacobsson@obgyn.gu.se; 7Department of Obstetrics and Gynecology, Sahlgrenska University Hospital, 413 45 Gothenburg, Sweden; 8Department of Infectious Diseases, Institute of Biomedicine, University of Gothenburg, 413 45 Gothenburg, Sweden; agnes.wold@microbio.gu.se

**Keywords:** umbilical, cord, plasma, parity, energy, amino acid, metabolism, metabolomics, arterial, venous

## Abstract

Umbilical cord blood is frequently used in health monitoring of the neonate. Results may be affected by the proportion of arterial and venous cord blood, the venous blood coming from the mother to supply oxygen and nutrients to the infant, and the arterial carrying waste products from the fetus. Here, we sampled arterial and venous umbilical cords separately from 48 newly delivered infants and examined plasma metabolomes using GC-MS/MS metabolomics. We investigated differences in metabolomes between arterial and venous blood and their associations with gestational length, birth weight, sex, and whether the baby was the first born or not, as well as maternal age and BMI. Using multilevel random forest analysis, a classification rate of 79% was achieved for arteriovenous differences (*p* = 0.004). Several monosaccharides had higher concentrations in the arterial cord plasma while amino acids were higher in venous plasma, suggesting that the main differences in the measured arterial and venous plasma metabolomes are related to amino acid and energy metabolism. Venous cord plasma metabolites related to energy metabolism were positively associated with parity (77% classification rate, *p* = 0.004) while arterial cord plasma metabolites were not. This underlines the importance of defining cord blood type for metabolomic studies.

## 1. Introduction

In the fetal circulation, blood circulates through the placenta to exchange gases, nutrients, and metabolites with the maternal blood through passive diffusion and active transport over the placental membranes [1,2]. The venous umbilical cord blood (from the placenta to the fetus) is rich in oxygen and supplies the fetus with nutrients necessary for survival and growth [3]. In contrast, the arterial umbilical blood (from the fetus to the placenta) transfers carbon dioxide and metabolic waste products back to the mother for excretion [4]. Modeling the difference in transport to and from the fetus can help to improve understanding of the interaction between the mother and the developing fetus. 

Umbilical cord blood is frequently used to monitor neonatal health. For clinical diagnosis purposes, umbilical cord blood may be sampled directly after delivery before the cord is severed (e.g., to screen arterial and venous pH as well as blood gas analysis for early signs of acidosis [5] and for collecting blood spots for diagnosis of inborn errors of metabolism). For research purposes, umbilical cord blood may be sampled to predict and assess neonatal health problems such as hypoxic-ischaemic encephalopathy [6], intrauterine growth restriction [7] and preeclampsia [8] or to prospectively predict childhood diseases (e.g., type 1 diabetes) [9]. Venous and arterial umbilical blood may be sampled separately using a syringe inserted into the respective blood vessel [7,10], but as this can be time consuming, frequently the blood is squeezed out from the cord after it is severed, resulting in the collection of a mixture of arterial and venous cord blood in unspecified proportions [11,12,13].

Previous studies have observed different levels of amino acids [14] as well as glucose, lactate, and catecholamines [15] between arterial and venous cord plasma and serum, but the differences have not been studied using a wider metabolic profiling approach, such as untargeted metabolomics, in which all measurable metabolites in a biological sample are detected (e.g., by GC-MS/MS).

Umbilical cord metabolites can be influenced by different factors such as gestational length [16] or birth weight [17,18,19,20,21,22], the infant’s sex [23], parity [24], and delivery mode [24] or maternal BMI [25,26]. We have previously looked into these traits using mixed arterial and venous blood plasma in another birth cohort and found significant differences in metabolome related to delivery mode, parity, and sex [27].

Our aim was to compare arterial and venous cord blood metabolomes. A secondary aim was to investigate associations between maternal and infant traits (gestational length, birth weight, sex, parity, maternal age, and BMI) using the arterial and venous cord plasma metabolomes. Our final aim was to replicate our previous findings from mixed umbilical cord blood in relation to sex and parity.

## 2. Results

Maternal and infant traits are reported in Table 1. In total, 416 features were used for statistical analysis, 30 from MRM, 117 from targeted and 269 from untargeted analysis (a complete list of annotated features can be found in Appendix A).

### 2.1. Differences in the Venous and Arterial Metabolome

The multilevel random forest model identified systematic differences between the venous and arterial metabolomes (Figure 1) (classification rate 79%, p_permutation_ = 0.004), and the metabolites selected by the model to be the most important for the classification are shown in Table 2. The amino acid, glutamic acid (FC_V/A_ 0.88), together with the TCA cycle intermediates α-ketoglutaric acid (FC_V/A_ 0.71) and succinic acid (FC_V/A_ 0.81), had higher concentrations in the venous cord plasma. In contrast, hydroxybutyric acid (FC_V/A_ 1.37), isoerythritol (FC_V/A_ 1.17), two deoxy-hexoses (FC_V/A_ 1.36 and 1.28), a hexose (FC_V/A_ 1.48), and hypoxanthine (FC_V/A_ 1.72), had higher concentrations in the arterial cord plasma.

### 2.2. Arterial and Venous Cord Plasma Metabolomes in Relation to Maternal and Infant Traits

Associations of the arterial and venous cord plasma metabolomes with the infant, maternal, and pregnancy traits (gestational length, birth weight, sex, parity, maternal age and BMI) that were investigated are shown in Table 3. Except for the association between venous cord plasma metabolome and parity (classification rate = 77%, p_permutation_ = 0.004), associations were null (Q2 ≈ 0 or classification rate ≈ 50%).

Most of the metabolites in venous cord plasma that associated with parity were related to energy metabolism and amino acid metabolism and were higher in the infants born to nulliparous mothers (Table 4). Although not all metabolites were univariately significant, they contributed to the multivariate model and are therefore included.

### 2.3. Replication Analysis

Several of the metabolites previously shown to be associated with parity in mixed arterial and venous cord plasma in another birth cohort [27] could be replicated in venous and arterial cord plasma separately in this study (significant associations in Table 5). However, none of the metabolites that were associated with sex in the previous birth cohort [27] could be replicated in the current study.

## 3. Discussion

In the present study, we observed differences in metabolite concentrations in arterial and venous cord plasma based on random forest analysis. Most of the metabolites that differed between venous and arterial cord blood plasma were related to energy metabolism (TCA-cycle intermediates) and amino acid metabolism. We found higher concentrations of glutamic acid in venous cord plasma, which suggests that this amino acid is transferred from the mother to the fetus. This finding is expected as the mother needs to supply the fetus with amino acids for protein synthesis, and glutamic acid is the most abundant in blood plasma. These results are supported by Holm et al., who found that most amino acids, measured by LCMS, were higher in venous compared to arterial cord plasma [14]. We also found higher levels of the TCA-cycle intermediates α-ketoglutaric acid and succinic acid in the venous plasma, indicating that the mother has an increased energy metabolism activity during labor, and therefore transfers high levels of these metabolites to the infant.

Arterial cord plasma instead contained higher concentrations of the purine base hypoxanthine, hydroxybutyric acid, a hexose, and two deoxy-hexoses and lower concentrations of the amino acid glutamic acid as well as the TCA cycle intermediates α -ketoglutaric acid and succinic acid compared to venous cord blood plasma. Our results contrast with a study by Koh et al., who, in a study of 57 infants born with elective caesarean section, found higher levels of glucose, lactate, and catecholamines in the venous cord plasma [15]. These observed discrepancies may thus be related to a difference in delivery mode, since only 5 of the 48 children were delivered with caesarean section (three elective and two acute) in our study. Thus, the vast majority of the women had vaginal deliveries. We hypothesize that differences in maternal energy expenditure between vaginal and elective caesarean delivery would lead to differences in what substrates are available in the maternal blood supply. Too few caesarean section deliveries were included in the studied group to enable meaningful analysis.

We further studied the relationship between the venous and arterial umbilical cord plasma metabolomes regarding several infant and maternal factors (i.e., gestational length, birth weight, sex, parity, maternal age, and maternal BMI). Of these factors, the only observable association was between parity and the venous cord plasma metabolome, indicating that the transfer of metabolites from the mother to the child during delivery differed depending on whether the mother has previously undergone labor or not. Almost all of the identified parity-associated metabolites were related to energy metabolism and amino acid metabolism and were higher in the venous cord plasma of nulliparous mothers. We speculate that these higher concentrations could relate to first deliveries being on average longer duration than subsequent deliveries [30], though the length of delivery was not recorded for this cohort. A longer time spent in labor could translate into an increased maternal and fetal energy requirement, supplied by the mother, and thus reflected in the venous cord plasma. We argue that since parity arguably relates more to the maternal than the fetal metabolism, venous cord plasma likely has more substantial potential to reflect parity. This could also explain why we were unable to find any solid multivariate associations of parity with the arterial cord plasma metabolome.

Parity is a factor that is often adjusted for when looking at neonatal and maternal health outcomes [31,32,33,34], as it may impact future health effects in both children and mothers. Parity has, for instance, been shown to associate with neonatal morbidity and perinatal mortality [35]. Parity has also been suggested to associate with cardiovascular disease risk in mothers [36]. Even though the implications of parity are relatively well studied, little is known about how parity exerts this effect [37]. Our study clearly shows that parity impacts the metabolites transferred from the maternal bloodstream to the fetus via the placental bloodstream at birth. Consequently, further investigation of metabolites as potential mediators could provide new insights into the influence of parity on health outcomes for mother and child.

Maternal BMI has previously been associated with hydroxybutyric acid [25], leucine, isoleucine [25,26], and several acetylcarnitines [26]. Also, increased birth weight has been associated with lysophosphatidylcholines [17,21,22], lysophosphatidylethanolamines [17], several amino acids [18,19], energy metabolism [20], tryptophan metabolism [18], acetylcarnitines [18,22], triglycerides [17,18], and nucleotide turnover-related metabolites [20]. Additionally, alanine, methionine, glycine, and tyrosine were previously shown to be higher in female than in male infants in dried blood spots taken between 48 h and 72 h after birth [23]. Conversely, acylcarnitines were higher in males [23]. The discrepancy between these findings and our largely null associations between venous and arterial cord metabolomes and maternal BMI or infant sex could be attributed to several factors: Previous studies have employed univariate modeling strategies, whereas we have employed a multivariate approach with stringent cross-validation to limit false-positive discovery. Differences in bioanalytical methodology could also explain some parts. The previous studies used targeted LCMS assays [18,21,25,26], NMR [19], untargeted LCMS metabolomics [20,22], and lipidomics [17] which all cover different parts of the metabolomes. Our use of GC-MS could lead to us missing certain associations found by others since the metabolite coverage is different between analytical methods. Further explanations for the discrepancies could also be attributed to differences in population, sample handling, and sample collection protocols.

Using univariate tests, we managed to replicate several of our previously reported associations regarding parity from another birth cohort that used the same GC-MS metabolomics methodology but mixed venous and arterial cord plasma [27]. In the present study, we found that several metabolites related to energy metabolism were similarly associated with parity in both cohorts. Surprisingly, several of these energy-related metabolites (e.g., malic acid and citric acid) associated with parity also in the arterial cord plasma even though the metabolomic profiles from this plasma did not associate with parity in the multivariate analysis. A possible reason for this could be that many of the associations are stronger in the venous plasma, thus making the weaker associations in the arterial plasma harder for the multivariate methods to detect. We were unable to replicate any of the associations previously found with regard to sex. These null findings could partly be explained by the fact that the associations found in our previous study in relation to sex were weaker than those of the findings related to parity, and therefore stand a larger chance of being spurious findings. Another reason could also be the low sample size both in this and the previous study, making weaker associations difficult to pick up. In addition, not all metabolites identified in the previous study could be found in the present data set.

A major strength of this study was the use of robust multivariate data analytical methodology. The repeated double cross-validation minimized false positive findings, and the multilevel approach managed to take the sample dependency into consideration. All samples were taken and processed in the same hospital, reducing variance in the sample management. The number of samples in this study is relatively small for the number of variables measured and should thus be interpreted with caution. A limitation of the study is a long time from the sampling of umbilical cord blood until centrifugation and subsequent freezing of some of the samples. Research personnel were only available during regular work hours, and samples from deliveries during other times were stored for up to two days in 4 °C before centrifugation. This may have led to differences in the rates of, for example, lipolysis and proteolysis, and is an inherent complication in collecting umbilical cord samples under clinical conditions. To estimate the effect of the error caused by the delay in freezing time we used the time until freezing as an independent variable and tried modeling this using the metabolome as dependent variables. However, the predictive performance of this model was low (Q2 = −0.08, *n* = 46), and we could therefore rule out systematic effects from the delay in time until freezing. However, the variability in time until freezing likely added non-systematic variability in the data as reviewed extensively by Stevens et al. [38]. We believe that the fact that our results are still significant after the induced variability from sub-optimal sample handling is a strong indicator that our findings are robust. This conclusion was further strengthened from sensitivity analyses performed only on samples with ≤24 h to get processed and frozen (*n* = 40). These analyses in fact showed even higher classification rate than our primary analyses (CR = 87.5% for arterial venous separation and CR = 82% for parity), suggesting a potential obfuscation of results from variability in the metabolome induced from sample management. Even though the sensitivity analyses resulted in higher prediction accuracy, the analysis performed on the larger study population both preserves statistical power and represents more conservative results and should therefore be considered the main analysis.

## 4. Materials and Methods

### 4.1. Study Subject and Sampling Protocol

In total, 51 matched venous and arterial cord blood samples were obtained at delivery from infants in the “Nutritional impact on Immunological maturation during Childhood in relation to the Environment” (NICE) cohort, a prospective birth-cohort consisting of children born at the Sunderby Hospital in northern Sweden between 2015 and 2018 [39]. Trained midwives sampled venous and arterial cord blood in connection with sampling for umbilical pH measurements. Blood was sampled using a syringe, prior to the cord being severed, and thereafter transferred to 500 µL EDTA tubes (Becton Dickinson, NJ, USA). Samples were stored at 4 °C at the maternity ward before being transported to the hospital laboratory the same or next working day (Monday–Friday). Upon arrival at the clinical laboratory, samples were centrifuged (5 min, 2400× *g*) and plasma aliquoted and stored in −80° freezers. Validation of correct sampling was performed by checking the pH of the arterial and venous umbilical cord blood, leading to removal of three samples where the umbilical artery sample had a higher or equal pH to that of the umbilical vein sample, making a total of 48 samples used for statistical analyses.

The NICE-study was approved by the Regional Ethical Review Board in Umeå (2013/18-31M, 2015-71-32) and written informed consent was obtained from the prospective parents during pregnancy for their own participation and after delivery for the participation of their child. The study was performed in accordance with the Declaration of Helsinki.

### 4.2. Sample Management

Samples were prepared for metabolomics analysis as described previously [40]. In brief, 100 µL plasma was thawed at 4 °C, and 900 µL of extraction solvent (containing 0.0625 ng/µL of all of the internal standards in MeOH:H_2_O (90:10 *v*/*v*)) was added (List of all internal standards can be found in Appendix A). The mixture was shaken on a bead shaker (30 Hz, 3 min), incubated (2 h, 4 °C), and centrifuged (17× *g*, 10 min). The supernatant (300 µL) was then transferred into GC-vials and evaporated to dryness in a MiVac vacuum evaporator system (Genevac, Ipswich, UK). For methoxymation, the samples were dissolved in 30 µL of methoxyamine in pyridine (20 mg/mL), incubated at 60 °C for 60 min, and stored overnight at room temperature. Samples were further derivatized by silylation by adding 30 µL of N-Methyl-N-(trimethylsilyl)trifluoroacetamide with 1% trimethylchlorosilane for one hour and finally, 30 µL heptane with 15 µL of 30 ng/µL methyl stearate (additional internal standard) was added before analysis by GC-MS/MS.

GC-MS/MS metabolomic analysis was performed by a combined scanning and multiple reaction monitoring (MRM) method [40] using a Shimadzu GCMS TQ-8030 GC-MS/MS system consisting of a Shimadzu GC-2010 Plus gas chromatograph, Shimadzu TQ-8030 triple quadrupole mass spectrometer, and a Shimadzu AOC-5000 Plus sample handling system (Shimadzu Europa GmbH, Duisburg, Germany). Data were acquired using the Shimadzu GCMS solutions software version 4.2. Samples (1 μL) were injected with a split ratio of 1:4 and analytes separated on a 15 m × 0.25 mm × 0.25 μm Rxi-5Sil MS column (Restek Corporation, Bellefonte, PA, USA). Injection port temperature was set to 270 °C, and the oven temperature program was as follows: 70 °C for 2 min and ramped to 330 °C at 30 °C/min where it was held for 2 min. GC was operated in constant linear velocity mode set to 80 cm/s. Septum purge flow was set to 3 mL/min. Interface and ion source temperatures were 290 and 200 °C, respectively. The autosampler was kept at 15 °C. Helium was used as the GC carrier gas, and argon was used as the MS/MS collision gas. MS data were collected using a combined scan and MRM method, scanning from *m*/*z* 50–750 over 75 ms and MRM transitions for 45 ms, for a total cycle time of 120 ms.

Feature extraction and metabolite identification from the GC-MS/MS was done by using three separate methods. (1) data from the pre-determined MRMs [40], (2) targeted extraction of metabolites from the scanning data by using a MatLab script [41] that extracts metabolites included in an inhouse library based on meeting criteria for retention index, full spectrum matching, and peak shape of a diagnostic *m*/*z* (analogous to SIM mode), provided by the Swedish Metabolomics Centre (all settings for targeted extraction available in Appendix A), and (3) untargeted metabolomics using MS-DIAL [42]. Key MS-DIAL settings were minimum, peak height: 50,000, peak width: 12 scans, sigma window: 0.75. Compounds were tentatively identified based on retention index and spectral matching against the combined GC-MS library available from MS-DIAL (GCMS DB_AllPublic-KovatsRI-VS2).

### 4.3. Statistical Analysis

Data from MRM, targeted, and untargeted data was combined after filtering by removal of identified peaks in untargeted data already present in targeted or MRM, and targeted data already present in MRM. Pre-processing and data analysis were conducted using R version 4.1.0 [43]. Imputation of missing values (0.026%) was performed using an in-house developed PLS-based imputation algorithm (function mvImpWrap; https://gitlab.com/CarlBrunius/StatTools/ (accessed on 11 December 2021)). Correction for inter- and intra-batch intensity variation was performed using the R package batchCorr [44].

All multivariate analyses were conducted with the MUVR algorithm [29], which uses random forest analysis in a repeated double cross-validation procedure, reducing the likelihood of overfitting and false-positive results. This algorithm also performs a minimally biased selection of the most informative variables (metabolites) of interest. To investigate differences between the arterial and venous plasma metabolomes, the MUVR random forest analyses were performed as multilevel analyses [28,29] to account for dependency for samples belonging to the same individual. In this analysis we used an effect matrix created from log2 fold change data calculated according to Equation (1), representing the within-individual variability between the arterial and venous metabolomes, separated from the between-individual variability.
(1)$$Effect\;matrix=\log_{2}\left(\frac{Arterial}{Venous}\right)$$

Permutation testing (*n* = 1000) was performed to ensure that results were not due to overfitting. Paired *t*-tests were performed on all reported metabolites from the MUVR algorithm, and Benjamini-Hochberg correction was applied to adjust for false discoveries [45]. Modeling of cord metabolome in relation to maternal and infant traits was performed similarly using the MUVR algorithm, using either the arterial or venous metabolomes as predictors and the traits as responses. Classification rates were calculated for the dichotomized predictors sex and parity whereas Q2 values were calculated for the continuous predictors. Models that were deemed to have potential predictive performance (defined *a priori* at Q2 > 0.2 or classification rate > 66%, arbitrarily set based on personal experience to achieve a prediction containing potentially meaningful information) were subjected to permutation testing (*n* = 1000). Metabolites of interest were extracted from models with p_permutation_ < 0.05 and were subjected to univariate testing using the Mann-Whitney U test.

Pathway analysis was attempted using MetaboAnalyst 5.0 [46]. However, the pathway coverage was deemed too low to give any additional information to the work (in total, 110 metabolites were overlapping between our annotated metabolites and those available in MetaboAnalyst, coverage for significant pathways after FDR correction was 2 out of 16 for the histidine metabolism pathway and 3 out of 21 for the beta-alanine pathway).

In addition, we tested if earlier findings from a previous study [27], in which we found several mixed cord blood metabolites to be associated with sex and parity, using the Mann Whitney U test with α = 0.05.

The datasets generated during and/or analyzed during the current study are available from the corresponding author on reasonable request and fulfilment of ethical requirements.

## 5. Conclusions

We observed systematic differences between the arterial and venous umbilical cord plasma metabolomes, primarily concerning energy metabolism. This suggests that the choice of arterial, venous, or mixed umbilical cord blood could have implications for research questions involving metabolic regulation, especially regarding energy and amino acid metabolism. Moreover, the use of mixed, arterial, or venous cord blood could contribute to difficulties when trying to compare studies, especially given the potential for differing proportions of arterial and venous blood in mixed cord blood samples. We further found that parity associated with metabolites in the venous umbilical cord plasma related to energy metabolism, meaning that whether the mother has undergone previous deliveries affects the umbilical cord plasma metabolome. This may be an important factor to account for when matching case-control pairs and for outcomes that may be related to energy metabolism. In contrast to the literature, we found neither arterial nor venous umbilical cord metabolomes to be associated with gestational age, gestational weight, sex, maternal age, or maternal BMI. However, the lack of associations could also be related to variability due to unavoidable pre-analytical sample management, the choice of metabolomics method, and sample size. Future studies on larger sample sets, with more uniform pre-analytical conditions and with analytical techniques providing wider coverage of the metabolome, are needed to better understand variability between the arterial and venous cord blood metabolome at birth.

## Figures and Tables

**Figure 1 metabolites-12-00175-f001:**
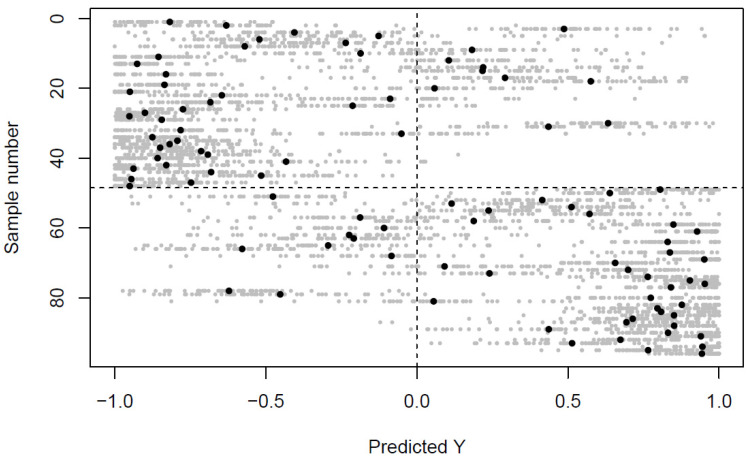
Multilevel classification of arterial versus venous cord blood. The effect matrix (sample numbers in the top half of the plot) and its negative counterpart (sample numbers in the bottom half of the plot) are modeled against a target vector of (−1) for the top half and (+1) for the bottom half of the samples [28,29]. Thus, the decision boundary for predictions is at zero, and observations ending up in the top left and bottom right corner are correctly classified. Predictions closer to the target values represent higher prediction accuracy. Small grey dots represent individual predictions at each repetition of the algorithm, whereas large black dots represent the averaged prediction across all repetitions.

**Table 1 metabolites-12-00175-t001:** Median values and IQR or quantity (*N*) and % for the infant and maternal traits, gestational length, birth weight, sex, parity, cesarean section, maternal age and, maternal BMI for all samples and separated for infants born to nulliparous and multiparous mothers.

	All Samples (*n* = 48)	Nulliparous (*n* = 23)	Multiparous (*n* = 25)
Gestational length (weeks + days)	40 + 0 (39 + 2–40 + 6)	40 + 1 (39 + 3–41 + 1)	39 + 6 (39 + 2–40 + 3)
Birth weight (g)	3510 (3310–3914)	3515 (3330–3778)	3505 (3310–3940)
Sex (*N* female)	25 (52%)	14 (61%)	11 (44%)
Parity (*N* nulliparous)	23 (48%)	23 (100%)	0 (0%)
Cesarean section (*N* yes)	5 (10%)	3 (13%)	2 (8%)
Age of mother (years)	28 (25–32)	28 (25–30)	29 (27–34)
Maternal BMI (kg/m^2^)	23.5 (22–27.1)	25.4 (23–28.9)	22.8 (20.8–25.7)

**Table 2 metabolites-12-00175-t002:** Metabolites of interest selected by the MUVR algorithm to differentiate between venous and arterial cord plasma, average fold change between arterial and venous plasma and *p*-values from Mann-Whitney U testing before and after Benjamini-Hochberg (FDR) adjustment for multiple comparisons.

Metabolites ^a^	FC ^b^	p_unadjusted_	p_FDR_	Metabolite Category
Higher levels in arterial cord plasma (blood from the fetus)				
Hypoxanthine	1.72	<0.001	<0.001	Purine base
Hexose	1.48	<0.001	<0.001	Sugar
177@1917 ^c^	1.46	<0.001	<0.001	Unknown
Hydroxybutyric acid	1.37	<0.001	<0.001	Glutathione-/fatty acid metabolism
Deoxy-hexose	1.36	<0.001	<0.001	Deoxy-sugar
Deoxy-hexose	1.28	<0.001	<0.001	Deoxy-sugar
72@1988	1.20	<0.001	<0.001	Unknown
Isoerythritol	1.17	<0.001	<0.001	Sugar alcohol
Higher levels in venous cord plasma (blood from the mother)				
α-ketoglutaric acid	0.71	<0.001	<0.001	TCA-cycle
Succinic acid	0.81	<0.001	<0.001	TCA-cycle
Glutamic acid	0.88	<0.001	<0.001	Amino acid

^a^ Metabolites selected from the MUVR-random forest algorithm. ^b^ Fold change of metabolite levels in arterial divided by metabolite levels in venous cord plasma, values > 1 indicating a higher concentration in the umbilical artery. ^c^ Metabolite could not be identified, reported as mass-to-charge ratio of main fragment @ Kovats retention index.

**Table 3 metabolites-12-00175-t003:** Associations between infant and maternal traits and the arterial and venous metabolomic profiles presented as % correctly classified samples for dichotomous variables and Q2 for continuous.

	Arterial	Venous
Gestational length (Q2)	0.11	−0.05
Birth weight (Q2)	−0.03	−0.12
Sex (CR ^a^)	56%	46%
Parity (CR ^a^)	62%	77% ^b^
Age of mother (Q2)	−0.00	−0.07
Maternal BMI (Q2)	−0.22	−0.10

^a^ CR, classification rate. ^b^
*p* = 0.004 (permutation test performed for this model only, since the other models did not meet criteria for predictive performance, i.e., Q2 > 0.2 or CR > 66%).

**Table 4 metabolites-12-00175-t004:** Metabolites selected by the MUVR algorithm to differentiate between infants born to nulliparous and multiparous mothers in venous cord blood together with *p*-values obtained from Mann-Whitney U testing before and after Benjamini-Hochberg (FDR) adjustment for multiple comparisons.

Metabolites ^a^	FC ^b^	p_unadjusted_	pFDR	Metabolite Category
Higher levels in infants with nulliparous mothers				
204@1879 ^c^	2.14	<0.001	<0.001	Unknown
89@1060 ^c^	1.98	<0.001	<0.001	Unknown
Pyruvic acid	1.95	<0.001	<0.001	Glycolysis
Histidine	1.88	<0.001	<0.001	Amino acid
Malic acid	1.78	<0.001	<0.001	TCA-cycle
Glucuronic acid	1.77	0.002	0.002	Carbohydrate conjugate
174@1877 ^c^	1.70	<0.001	<0.001	Unknown
Sarcosine	1.65	<0.001	<0.001	Amino acid metabolism
Oxalic acid	1.59	<0.001	<0.001	TCA-cycle related
Isocitric acid	1.48	<0.001	<0.001	TCA-cycle
52.05@1106 ^c^	1.28	<0.001	<0.001	Unknown
Nicotinic acid	1.26	<0.001	<0.001	Vitamin B3
73@1861 ^c^	1.20	0.003	0.003	Unknown
Higher levels in infants with multiparous mothers				
Aminobutyric acid	0.84	0.13	0.13	Amino acid metabolism

^a^ Metabolites selected from the MUVR-random forest algorithm. ^b^ Fold change of metabolite levels in parous divided by metabolite levels in nulliparous cord plasma, values > 1 indicate a higher concentration in the venous cord blood of infants born to nulliparous mothers. ^c^ Metabolite could not be identified, reported as mass-to-charge ratio of main fragment @ Kovats retention index.

**Table 5 metabolites-12-00175-t005:** Replicated associations of previously reported metabolites [27] with parity.

	Venous		Arterial	
Metabolite	FC ^a^	pFDR	FC ^a^	pFDR
N-acetyl mannosamine	1.16	0.109	1.16	0.041
Isocitric acid	1.45	<0.001	1.32	0.016
Sorbitol	1.39	0.109	1.69	0.029
Malic acid	1.79	<0.001	1.22	0.060
Lactulose	1.47	0.020	1.37	0.029
Citric acid	1.85	0.039	1.89	0.024

^a^ Values > 1 for fold change indicate higher levels in infants born to nulliparous mothers.

## Data Availability

The data presented in this study are available on request from the corresponding author. The data are not publicly available due to ethical restrictions.

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
