# Peer review of "Differences between Arterial and Venous Umbilical Cord Plasma Metabolome and Association with Parity"

_metabolites, 2022, doi:10.3390/metabo12020175_

Round 1
Reviewer 1 Report
The manuscript “Differences between arterial and venous umbilical cord plasma metabolome and association with parity” describes a comparative study between the GC-MS metabolomes of arterial and venous umbilical cord blood (n=48), and their relationships with gestational length, birth weight, sex, parity, maternal age, and BMI. This topic is important and interested to readers, but the research design has serious flaws. Major concerns are:
- Blood samples were centrifuged after a long time, on the next working day in some cases as referred in Material and Methods (lines 247-251). Despite authors were aware of this issue, as referred in discussion (lines 229-239), pre-centrifugation processing delays may influence metabolite levels and introduce inter-subject variability. As reviewed by Stevens et al. (doi: 10.3390/metabo9080156), studies investigating the blood processing times revealed that energy-related metabolites are the most sensitive to time delays. Hence, the differences obtained in energy-related metabolites between arterial and venous plasma metabolomes reported in this study may have a high degree of uncertainty.
- In results (lines 78-79), peaks from three different approaches are described, namely MRM, targeted and untargeted, but these analyses were not properly described in Materials and Methods.
- Pre-processing parameters used for the three analyses were not described. In addition, no clear discrimination between arterial and venous plasma was shown by the multivariate analysis methods employed and a low predictive performance (Q2 > 0.2 or classification rate > 66%) was defined without a reasonably justification. The number of permutations used was very low (n=100).
- A table describing the characteristics of samples with the definition of the number of samples in each group is missing (e.g., number of first-born and non-first-born children, etc).
Author Response
- We agree that this is indeed a major issue with the study design. However, it is very difficult to achieve perfectly homogenous sample handling when research personnel can only be available during normal working hours. We have now put more emphasis on this issue in the limitations section along with adding the proposed reference to this section (Lines 244-260).
The section now reads:
“To estimate the effect of the error caused by the delay in freezing time we used the time until freezing as an independent variable and tried modeling this using the metabolome as dependent variables. However, the predictive performance of this model was low (Q2 = -0.08, n=46), and we could therefore disregard any systematic effect from the freezing time. However, the variability in time until freezing likely added non-systematic variability in the data as reviewed extensively by Stevens et al. [38]. We believe that the fact that our results are still significant after the induced variability from sub-optimal sample handling is a strong indicator that our findings are robust.”
We also believe that this issue would dilute any associations rather than inducing spurious findings, indicating that our results likely would have been even more pronounced with more uniform sampling. - Thank you for this comment, we have now tried to clarify this in the methods section by changing the relevant materials and methods section to read:
“Feature extraction and metabolite identification from the GC-MS/MS was done by using three separate methods. 1) data from the pre-determined MRMs [40], 2) targeted extraction of metabolites from the scanning data by using a MatLab script [41] that extracts metabolites included in an inhouse library based on meeting criteria for retention index, full spectrum matching, and peak shape of a diagnostic m/z (analogous to SIM mode), provided by the Swedish Metabolomics Centre (all settings for targeted extraction are available in Supplementary Material Suppl. Table S3), and 3) untargeted metabolomics using MS-DIAL [42].” Lines (312 to 319). - The only pre-processing treatment the data underwent was intensity normalization from instrument drift, using the batchCorr package. As both random forest and the Mann-Whitney U-test are scale invariant, no additional treatment of data was deemed necessary, to clarify this, the sentence on lines 329-330 were changed to now read:
“Correction for inter- and intra-batch intensity variation was performed using the R package batchCorr [44].”.
We would also like to disagree with there not being any clear discrimination between the arterial and venous plasma, as 79 % classification rate is actually quite high. We agree that the a priori limits for Q2 and classification rates are indeed arbitrary: They are in fact set from previous experience working with the MUVR algorithm to a level where meaningful information can be gained from a model, where Q2>0.2 in most cases represent an observable correlation and a classification rate of >66% indicating more than two thirds of the samples being able to be classified correctly. An elaboration of the limits are now found on lines 350-353 now reading:
“Models that were deemed to have potential predictive performance (defined a priori at Q2 > 0.2 or classification rate > 66 %, arbitrarily set based on personal experience to achieve a prediction containing potentially meaningful information) were subjected to permutation testing (n=1000).”
100 permutations were initially deemed sufficient, based on visual inspection of the distribution of Q2 values from the permutation testing, but have now been increased to 1000. - Thank you for this point. We have now added a table showing this information separated for children born to nulliparous and multiparous women. In addition we also added further information of delivery mode. See Table 1 revised manuscript line 79
Reviewer 2 Report
The current study by Hartvigsson and coworkers explores the differences between arterial and venous umbilical cord plasma metabolome. To this end, the study team uses metabolomics data obtained from 51 clinical samples (venous and arterial umbilical blood) in conjunction with random forest analyses. Conceptually, it is an interesting study combining state-of-the-art metabolomics with unbiased statistical analyses. The potential of this study is to identify metabolites with predictive values related to inherited disorders or pregnancy complications. Despite its potential, the study in its current form has several caveats and limitations. 1. The sample size is small, thus, the robustness of analyses is limited. 2. The authors acknowledge that the handling of samples was not consistent. Sample integrity and quality are of utmost importance in metabolomics studies since metabolic intermediates are unstable and decompose fast if the biological material is not frozen.
Overall, the study team needs to address adequately the aforementioned comments
Author Response
- Thank you for this comment. We agree that this is indeed a limitation of our study, but we still believe that our results are of value despite the low sample size as we are not aware of any other study looking into these factors are using a larger sample size. Further discussion about this limitation has been added in lines 237-238 now reading:
“The number of samples in this study is relatively small for the number of variables measured and should thus be interpreted with caution.”. - We appreciate that the reviewer notes that we have acknowledged this issue. We realize that this is a weakness of the study design, but we believe that this weakness is more likely to dilute our results than it is to cause spurious findings in our data. We have added further discussion about this issue on lines 238-260, now reading:
“A limitation of the study is a long time from the sampling of umbilical cord blood until centrifugation and subsequent freezing of some of the samples: research personnel were only available during regular work hours, and samples from deliveries during other times were stored for up to two days in 4 °C before centrifugation. This may have led to differences in the rates of, e.g., lipolysis and proteolysis, and is an inherent complication in collecting umbilical cord samples under clinical conditions. To estimate the effect of the error caused by the delay in freezing time we used the time until freezing as an independent variable and tried modeling this using the metabolome as dependent variables. However, the predictive performance of this model was low (Q2 = -0.08, n=46), and we could therefore rule out systematic effects from the delay in tie until freezing. However, the variability in time until freezing likely added non-systematic variability in the data as reviewed extensively by Stevens et al. [38]. We believe that the fact that our results are still significant after the induced variability from sub-optimal sample handling is a strong indicator that our findings are robust. This conclusion was further strengthened from sensitivity analyses performed only on samples with ≤ 24 h to get processed and frozen (n = 40). These analyses in fact showed even higher classification rate than our primary analyses (CR = 87.5 % for arterial venous separation and CR = 82 % for parity), suggesting a potential obfuscation of results from variability in the metabolome induced from sample management. Even though the sensitivity analyses resulted in higher prediction accuracy, the analysis performed on the larger study population both preserves statistical power and represents more conservative results and should therefore be considered the main analysis.”
Reviewer 3 Report
The study titled “Differences between arterial and venous umbilical cord plasma metabolome and association with parity” investigated differences between arterial and venous cord plasma metabolomes suggesting links of the latter with parity. The study suggests differences in energy metabolism of first-time mothers versus those that have had previous deliveries, and this may add to the current knowledge on maternal or neonatal health outcomes associated with parity.
Major Comments
Methods
The authors need to divulge more details around the sampling for the cord blood. Especially considering that the blood was collected over a 3-yr duration from the study cohort, how was this done in a consistent manner? Was it immediately after delivery and cord clamping?
Anthropometric and basic clinical characteristics of mothers and infants of the study cohort have not been provided in detail. I suggest these be tabulated instead of the short paragraph at the beginning of the results section (Line 73-77). Again, these should be available as supplementary if not in the main body of text.
As highlighted by the authors, the fact that the cord blood was stored at refrigeration temperature for 24/48 hours is not ideal and might explain why some of their results differ from other studies in the literature. Since the authors have taken this limitation into account and claimed that there was no systematic effect because of freezing delays (line 237), that analysis should be presented as supplementary data.
Internal standards (line 260) need to be stated.
How was the data normalised? Using internal standard normalisation? Please add this in the methods section.
Minor comments
Line 301-304- should be mentioned in the sampling protocol
Line 328- Why was pathway coverage low? Exactly how many metabolites were used for this analysis? Which program was used for this? Pathway analysis for this type of study would be very informative.
Line 333-All identified metabolites should be stated in supplementary
Results
Line 84-87 Fold changes should be provided here for the metabolites being discussed
There are two table 3’s in the manuscript. The last one should be Table 4
Line 138-Why could they not be replicated?
176-Although the hypothesis proposed seems logical, it seems largely speculative without any data provided around the length of deliveries. Have all other possible reasonings been considered from the literature?
Author Response
1. The authors need to divulge more details around the sampling for the cord blood. Especially considering that the blood was collected over a 3-yr duration from the study cohort, how was this done in a consistent manner? Was it immediately after delivery and cord clamping?
Thank you for this comment, which highlighted that this was not described in enough detail. Further information has been added to the lines 269-270 now reading:
“Blood was sampled using a syringe, prior to the cord being severed, and thereafter transferred to 500 µl EDTA tubes”
2. Anthropometric and basic clinical characteristics of mothers and infants of the study cohort have not been provided in detail. I suggest these be tabulated instead of the short paragraph at the beginning of the results section (Line 73-77). Again, these should be available as supplementary if not in the main body of text.
Thank you for this comment, a table has now been made instead of the text paragraph. In addition we have also added separate columns for children born to nulliparous and multiparous women and added information about delivery mode. See Table 1 (Line 79) in the revised manuscript.
3. As highlighted by the authors, the fact that the cord blood was stored at refrigeration temperature for 24/48 hours is not ideal and might explain why some of their results differ from other studies in the literature. Since the authors have taken this limitation into account and claimed that there was no systematic effect because of freezing delays (line 237), that analysis should be presented as supplementary data.
We agree with the reviewer that this indeed constitutes sub-optimal sample handling. We believe that this might have diluted our results causing them to be weaker than they would have been if the samples were handled optimally. We still, however, believe that our data makes a valuable contribution to the research field. More details regarding the analysis of the freezing details along with further elaboration of this issue have been added into the text at lines 242-260 now reading:
“This may have led to differences in the rates of, e.g., lipolysis and proteolysis, and is an inherent complication in collecting umbilical cord samples under clinical conditions. To estimate the effect of the error caused by the delay in freezing time we used the time until freezing as an independent variable and tried modeling this using the metabolome as dependent variables. However, the predictive performance of this model was low (Q2 = -0.08, n=46), and we could therefore rule out systematic effects from the delay in tie until freezing. However, the variability in time until freezing likely added non-systematic variability in the data as reviewed extensively by Stevens et al. [38]. We believe that the fact that our results are still significant after the induced variability from sub-optimal sample handling is a strong indicator that our findings are robust. This conclusion was further strengthened from sensitivity analyses performed only on samples with ≤ 24 h to get processed and frozen (n = 40). These analyses in fact showed even higher classification rate than our primary analyses (CR = 87.5 % for arterial venous separation and CR = 82 % for parity), suggesting a potential obfuscation of results from variability in the metabolome induced from sample management. Even though the sensitivity analyses resulted in higher prediction accuracy, the analysis performed on the larger study population both preserves statistical power and represents more conservative results and should therefore be considered the main analysis.”
4. Internal standards (line 260) need to be stated.
We thank the reviewer for this comment. Internal standards are now available in the supplementary materials Suppl Table S2.
5. How was the data normalised? Using internal standard normalisation? Please add this in the methods section.
The only pre-processing treatment the data underwent was the intensity normalization using the batchCorr package. As both random forest and the Mann-Whitney U-test are scale invariant, no additional treatment of data was deemed necessary, to clarify this, the sentence on lines 329-330 were changed to now read:
“Correction for inter- and intra-batch intensity variation was performed using the R package batchCorr [44].”.
Minor comments
6. Line 301-304- should be mentioned in the sampling protocol
Thank you for this comment. The suggested lines have now been moved earlier in the manuscript and can now be read in lines 274-277.
7. Line 328- Why was pathway coverage low? Exactly how many metabolites were used for this analysis? Which program was used for this? Pathway analysis for this type of study would be very informative.
Pathway coverage was likely too low due to not enough annotated metabolites were found. The requested information on the pathway analysis has now been added to lines 346-351 reading: “Pathway analysis was attempted using MetaboAnalyst 5.0 [46]. However, the pathway coverage was deemed too low to give any additional information to the work (in total, 110 metabolites were overlapping between our annotated metabolites and those available in MetaboAnalyst, coverage for significant pathways after FDR correction was 2 out of 16 for the histidine metabolism pathway and 3 out of 21 for the beta-alanine metabolism pathway).”
8. Line 333-All identified metabolites should be stated in supplementary
Thank you for this comment, a list of annotated metabolites is now available in Supplementary table S1.
Results
9. Line 84-87 Fold changes should be provided here for the metabolites being discussed
Thank you, fold changes are now added. (Lines 83-90)
10. There are two table 3’s in the manuscript. The last one should be Table 4
We thank the reviewer for spotting this error, it has now been corrected.
11. Line 138-Why could they not be replicated?
Excellent question, we realize now that an explanation of this was omitted. Speculation around this has now been added to the discussion section lines 225-232 now reading:
” We were unable to replicate any of the associations previously found with regard to sex. These null findings could partly be explained by the fact that the associations found in our previous study in relation to sex were weaker than those of the findings related to parity, and therefore stand a larger chance of being spurious findings. Another reason could also be the low sample size both in this and the previous study, making weaker associations difficult to pick up. In addition, not all metabolites identified in the previous study could be found in the present data set.”
12. 176-Although the hypothesis proposed seems logical, it seems largely speculative without any data provided around the length of deliveries. Have all other possible reasonings been considered from the literature?
Thank you for this important point. We agree that this is indeed problematic, and that we cannot say anything for certain. The word hypothesize has now been changed to speculate in the text (Line 179) to reflect this. We have no other explanation of this from the literature.
Round 2
Reviewer 1 Report
Authors addressed the major concerns and the manuscript was significantly improved.
Reviewer 2 Report
The authors addressed the reviewer's requests. The manuscript is significantly improved.
Reviewer 3 Report
Thank you for submitting the rebuttal and the revised manuscript.
Although I am a bit concerned that your previous findings have not been replicated in this study, I am satisfied with the acknowledgement about the same. Also whilst I believe that sample collection has a serious limitation, I also agree that obtaining cord samples in a consistent manner is a real challenge!
Overall I am satisfied with the revisions.